# Building health: Motivational factors to enhance systematic work environment management in the construction industry

Niklas Rydbo[1]*, Kristina Aurelius[2,3], Martin Grill[1], Christian Jacobsson[1], Anders Pousette[1]

1 Department of Psychology, University of Gothenburg, Gothenburg, Sweden, 2 Occupational and Environmental Medicine, School of Public Health and Community Medicine, Sahlgrenska Academy, University of Gothenburg, Sweden, 3 Occupational and Environmental Medicine, Sahlgrenska University Hospital, Region Västra Götaland, Sweden

☯ These authors contributed equally to this work.

* niklas.rydbo@psy.gu.se

## Abstract

### Objective

The aim of the study was to investigate whether and how the interventions have been beneficial in developing a well-functioning systematic OSH management. The aim was also to investigate whether and how the interventions have facilitated or impeded autonomous motivation to develop a well-functioning systematic OSH management.

### Method

Data were collected through individual interviews with participants in the Health Construction intervention project and audio recordings from the concluding workshop for each batch. The data were analyzed using an inductive thematic analysis.

### Results

The seven themes from the analysis yielded the following results. Participation in the project was highlighted as a key factor for development, as it offered opportunities to collaborate and share experiences and insights about the domain of the occupational environment with representatives from other companies. Establishing a development team within participating companies appears to have influenced the legitimacy of occupational safety and health, as well as the assigned safety priority. A process consultative approach by external consultants seems to have contributed to an enhanced ability to articulate thoughts, identify solutions, synthesize discussions, and work effectively within the development team. The results suggest that the assigned safety priority of the respective participating companies has increased. Additionally, there are indications that autonomous motivation to continue and deepen efforts to address

**Data availability statement:** Data cannot be shared publicly because ethics approval (Dnr 1143-17) from the Regional Ethical Review Board in Gothenburg was granted on the condition that results are presented only at an aggregate level and in a way that does not identify individuals. However, anonymised data are available upon request For data requests, contact Niklas Rydbo, Department of Psychology, University of Gothenburg, (niklas.rydbo@psy.gu.se) or Andreas Segerberg, the institutional representative for data storage at the Department of Psychology, University of Gothenburg (andreas.segerberg@psy.gu.se).

**Funding:** This work was supported by AFA Insurance (grant number 160151 ) and Swedish Transport Administration (grant number TRV2016/107905) and Region Västra Götaland, Occupational and Environmental Medicine (in-kind financing). The funders had no role in study design, data collection and analysis, decision to publish, or preparation of the manuscript.

**Competing interests:** The authors have declared that no competing interests exist.

safety and health issues in a structured way has increased within the participating companies.

## Conclusions

The results of this study indicate that the interventions implemented in the project have been beneficial in fostering the development of a well-functioning systematic occupational safety and health management. This may have been achieved by enhancing the autonomous motivation to prioritize and develop a healthy and safe work environment within the participating companies.

## Introduction

The construction industry has more and more considerable potential health and safety risks than in most other sectors, and it is one of the most dangerous industries in terms of occupational safety and health (OSH), both in Europe [1,2] and worldwide [3,4]. The European construction industry has one of the highest numbers of workers reporting exposure that affects their physical health, with risk factors including strenuous work postures, repetitive work movements, handling heavy loads, noise, vibration, smoke, chemicals, and dust. Exposure affecting mental health is reported considerably less often than physical health [1]. In Sweden, occupational diseases in the construction industry are primarily characterized by strain injuries, which account for approximately 60% of the reported cases. Physical factors, including noise and vibration, account for around 25% [5]. About one in three persons employed in the sector has some form of work-related disability, which is considered exceptionally high [6]. Previous literature indicates that there are significant challenges in implementing effective OSH practices in small and medium-sized enterprises (SMEs) [7–11], that is, companies with fewer than 250 employees [12,13].

SMEs account for approximately two-thirds of employment in the EU, are present in all industries [11], and are particularly prevalent in the construction industry [14]. SMEs appear to have more difficulties establishing systematic ways to manage OSH than larger companies [13], and the possibilities to develop effective OSH management increase with company size [11,15]. Large construction companies often have expertise that works systematically to create a safe and healthy environment [11,16]. In Sweden, national legislation requires employers to integrate an OSH management system, known as Systematic Work Environment Management, into their daily operations [17]. This OSH management system complies with internationally implemented legislation providing a safe working environment [18]. The Swedish Work Environment Authority provides rules, regulations, and tools [17] to support OSH management. However, according to the Swedish Work Environment Authority, establishing an effective and functioning OSH management system in SMEs is problematic [15,19], and the success of regulatory strategies and campaigns aimed at addressing OSH challenges is limited [8,20,21]. In a review of 267 studies and reports, Frick and Johansson [19] noted that more knowledge is needed about the development

processes that lead to effective OSH management. The findings suggest that research in this area is still limited and that more research is required to understand what works for whom and in what context [8,22,23]

Based on the work environment situation in SMEs in the construction industry, the intervention project Building Health (Hälsobygget) [24–26] was designed. The project aimed to develop and test a methodology for achieving a well-functioning and continuous work environment management in smaller construction companies within the supply chains of large construction projects. The design of the project's interventions was based on previous research that demonstrated employee involvement in work environment management and the adaptation of OSH management to specific company conditions as essential factors in small companies that had successfully developed effective OSH management [16]. The design was also based on the so-called network model, where different companies collaborate to establish effective OSH management within their own company [27]. To complement the network model, support for each company through expert advice and process consultation [28] was added.

This study is about the intervention project Building Health and the aspects of the projects interventions that the participants perceived as conducive to the development of effective OSH management

## Starting points for the project

**The points on which the design of the interventions was based were.** 1) Assigned safety priority [29,30], 2) Motivation to improve work environment according to Self Determination Theory (SDT) [31] and the development of collective efficacy [32], 3) Involvement of first-line managers together with employees [33,34], 4) A process consultative approach to identify workplace problems, taking responsibility for issues and finding solutions [28] combined with a model for motivational communication [35], and 5) Problem-based learning (PBL) [36,37]. The starting points will be described in more detail in the following section.

**Assigned safety priority.** Assigned safety priority has been identified as one of the four core safety climate dimensions [29]. It can be defined as managers expressed concern for employees' health and safety [30]. How employees perceive the safety climate relates to how managers act regarding employees' well-being [30]. When meeting production targets is prioritized over safety across time and situations, production is viewed as a priority. Employees will then align their behavior with the perceived priorities, often at the expense of safety [38]. On the other hand, when managers express a commitment to safety and continually demonstrate through words and actions that they care about the well-being and safety of employees, especially when safety conflicts with production goals, it promotes a common perception among employees that safety is a high priority, leading to a better safety climate [30,38]. The composition of the development team (hereinafter referred to as "the team") described below was based on the importance of expressing the importance of working with OSH management within the participating companies. The fact that the participating companies were part of a supply chain were also benefited by i) the client chose to compensate the participating companies financially for the costs of participating in the project and ii) the participation of the client and the main contractor in the first workshop where they were invited to share their views on OSH and express their opinions regarding the importance of effective OSH management in the infrastructure project.

**Motivation to improve the work environment and the development of collective efficacy.** Previous research literature has indicated that commitment to safety behavior is influenced by employee's safety motivation [39,40]. The selection of the design for the intervention structure was based on the assumption that modifying an existing network model [27] within the project would foster collective efficacy [39] – defined as the collective capacity within an organization to undertake and complete a developmental task – as well as autonomous motivation thru the three fundamental psychological needs outlined in the motivation theory Self-Determination Theory (SDT) [41–44]. Previous research has indicated a positive correlation between collective efficacy and various forms of motivation [45–47] and that the perception of collective efficacy, in this case, creating a good work environment, motivates groups toward the task at hand [32]. The concept has also been identified as crucial when developing OSH interventions [48,49]. When coupled with the perception

of competence, a sense of belonging, and autonomy in SDT, it fosters autonomous motivation [41,42] for the task and enhances the group's resilience to adversity. Previous research in the field of OSH indicates that motivation is a significant factor influencing both safety compliance and safety participation [40,50,51]. Nykänen et al. [51] identify a research gap related to the specific mechanisms of safety motivation interventions. They emphasize the need for further research to understand what factors trigger motivation in order to develop effective intervention strategies.

**Involvement of first-line managers together with employees.** Previous research has identified employee participation and involvement as pivotal to the efficacy of occupational OSH interventions [33,34], and the collaboration between employers and employees has been highlighted as essential in designing interventions [33,50]. A steering group responsible for implementing OSH intervention programs needs to comprise employer representatives and employees [34]. The participating companies were each represented by a development team, appointed by the respective company's management. It was the ambition that the team would consist of a top-level manager of the company, a first-line manager/supervisor, and two employee representatives, at least one of whom was a safety representative. The team participated in each workshop and was responsible for the OSH work selected by their own company.

**Process consultation and motivational interviewing.** The consultation approach from both experts and process consultants was based on the idea of process consultation by Schein [52,53], where the central assumption is that "one can only help a human system to help itself" [52, p.1] and that the client and the consultant, as equal partners, share responsibility for the desired outcome [54]. When both consultant and client see each other as partners in the change and its consequences, the risk of applying interventions that are not adapted to the needs and challenges of the organization is reduced. The consultant takes on the role of a facilitator and tries to satisfy clients' needs by helping them find the answers rather than providing them, thus making the client autonomous and independent [54]. The process and expert consultants were also introduced to Motivational Interviewing, an evidence-based practice that focuses on creating collaborative conversations about change to strengthen a person's reasons, motivation, and commitment to change [35]. The three basic psychological needs [42] that are articulated in SDT are all directly addressed in motivational interviewing [55], and one key element in the spirit of motivational interviewing is the support of autonomy [55,56]. The role of the counselor is to respect the client's autonomy, help the client explore capacities and reasons for change [57], evoke motivation, and assist the client in making decisions [35]. In addition, advice and information are provided only with the client's permission [35].

**Problem-based learning.** Problem-based learning (PBL) [36] is, from the beginning, a student-centered pedagogical approach most deployed in higher education. Even though there are various definitions of PBL, a common and central premise is that a learning process is initiated by a problem [37]. PBL has been applied in different contexts. Previous research has identified the potential of a work-based PBL model (wPBL) to address company problems and facilitate continuous learning in SMEs [58]. In the project, the teams were considered PBL groups. The interventions conducted during the workshops—whether within the team, across groups, in professional groups, or with the entire group—were based on the steps outlined in the most widely used structure of PBL. [58]. Step 1: Clarify the concepts in question. Identifying and clarifying any unfamiliar terminology included in the problem description is essential. The process ensures that the problem is fully comprehended.

Step 2. Define the problem. The problem is defined through the formulation of questions and investigations.

Step 3. Engage in a brainstorming session. Members identify potential hypotheses and solutions based on their existing knowledge, utilizing the group's collective knowledge to identify areas of uncertainty.

Step 4. Evaluate the proposed solutions. Review steps two and three to evaluate the potential solutions. Select a suitable solution.

Step 5. Synthesis. The groups share results, experiences, and lessons learned, both within and between them.

As a model for continuous improvement [59,60] of the work environment, the Swedish OSH management wheel [17,61] was integrated into the PBL structure. It includes investigations, risk assessments, measures, and follow-ups linked to all working conditions.

                                                                                           

## Project implementation

Each project batch was conducted over a 6–8-month period and began with an initial full-day workshop. This was followed by five half-day workshop at approximately one-month intervals, including the final workshop. The purpose of the first workshop was to a) express the project's priority from the client, b) create a trusting relationship between the participants and the researchers with expertise in different areas, and c) take the first step in analyzing the work environment problems of the company. The purpose of workshop 2 was to start the problem analysis, clarify psychological contractual barriers, and support the action work. The summary from workshop 2 formed the basis for a company-specific plan that covered the specific problem the company wanted to focus on and outlined expectations for each party (employer/employee) relevant to the company and the specific problem. During workshop 3–5, concrete plans were developed for what would be achieved during the project period, what would be done for the next workshop, and who would be responsible for the different activities. Throughout all workshops, educational components were integrated with reflection and practical work on company-specific issues. Process consultants and experts provided support. At the last workshop, companies presented successful cases, and all participants, including the process consultants, shared experiences and lessons learned for the future. Between the workshop, companies had the opportunity to receive advice from subject matter experts in noise, ergonomics, chemical/physical exposure, and safety/psychosocial work environments, as well as support from the process consultant in their work.

## Aim of the study

The aim of the study was to investigate whether and how the interventions have been beneficial in developing a well-functioning systematic OSH management. The aim was also to investigate whether and how the interventions have facilitated or impeded autonomous motivation to develop a well-functioning systematic OSH management.

## Method

### Informants

A total of twenty-five individuals who had engaged in a development team under the aforementioned project were invited to participate in the interviews. Nineteen participants consented to participate and were interviewed. Table 1 provides further information on the number of participants from each company.

### Procedure

Nineteen individual semi-structured interviews were conducted via digital tools 1–4 months after the last workshop of each batch in the project. The interviews were conducted by four researchers from the project, all licensed psychologists with experience in occupational health research, from November 20, 2019, to August 13, 2021. The questions were guided by earlier studies in the field of evaluating OSH interventions [62,63] and formulated to elicit information about the experiences associated with participation in the project, including the development, fulfillment of objectives, the company's involvement during the project, the availability of resources, and the motivation to continue working with OSH management. Interviews were recorded and transcribed verbatim. A recording was made at the last workshop of each batch, where each company presented what they had achieved during the project. This recording was transcribed verbatim.

### Data analysis

An inductive data-driven thematic analysis was selected as the methodological approach. This decision was based on previous literature that underscored thematic analysis as an appropriate method for identifying themes and patterns of meaning based on current data in relation to the research question at hand [64,65]. The inductive approach was selected over the theoretical thematic analysis because the objective was to create a bottom-up analysis, i.e., the themes would

**Table 1. Informants from Building Health.**

| Company | Interviews | Objective during the Project |
|---|---|---|
| A | 2 Interviews<br>• 1 Human Resource Manager<br>• 1 Work Environment Coordinator | Keeping special equipment stores tidy and improving communication |
| B | 4 Interviews<br>• 2 Managers<br>• 1 Work Environment Coordinator<br>• 1 Skilled Worker | Reducing dust and dust-related risks, improving communication |
| C | 5 Interviews<br>• 1 Managing Director<br>• 1 Regional Director<br>• 3 Skilled Workers | Implementing systematic OSH management |
| D | 1 Interviews<br>• 1 Work Environment Coordinator | Improved occupational safety |
| E | 4 Interviews<br>• 1 Manager<br>• 3 Skilled Workers | Reducing threats and violence and helping employees deal with threats and violence |
| F | 3 Interviews<br>• 1 Business Area Manager<br>• 1 Human Resource Manager<br>• 1 Transport Manager | Reducing driver stress and improving communication. Improved occupational safety |

not be derived from existing theoretical frameworks [65]. Inductive analysis appears to offer a more comprehensive interpretation of data compared to deductive analysis [64].

The inductive thematic analysis [64,66] of the transcripts was conducted using Atlas.ti 21 software to identify essential processes on both individual and organizational levels.

In accordance with the phases described by Braun and Clarke [64], two of the authors initiated the first phase by becoming acquainted with the data. This was accomplished through a systematic examination of the transcripts, which involved repeated readings, generating preliminary ideas and initial codes that could capture salient features of the data. The second step in the analysis process, generating initial codes, involved the systematic coding of salient features of the data across the entire data set by one of the authors. This process entailed the aggregation of data relevant to each code. In the third phase of the analysis, the identification of themes was initiated through the systematic aggregation of codes by the coding author into potential themes encompassing all data relevant to each potential theme. All potential themes were provided with preliminary definitions. As a fourth step, all authors read and discussed all codes, preliminary subthemes, and preliminary themes to further identify meaningful themes and thematic structures. In step five, the coding author, based on the process in step four, defined and named the themes and subthemes [64,66,67], which were further processed, reviewed, and refined until the final list of themes was agreed upon by all the authors.

## Ethics

Ethics approval (Dnr 1143−17) was obtained from the Regional Ethical Review Board in Gothenburg, as this research involved human participation. The participants provided informed verbal consent during the interviews, which were recorded and transcribed verbatim. All participants in the project provided written informed consent for the utilization of all material, in both written and recorded formats, derived from workshops.

## Results

The analysis if the data resulted in the identification of seven main themes with no sub-themes, outlined in Table 2.

**Table 2. Themes.**

| |
|---|
| • OSH on the agenda |
| • The development team drives development |
| • Expert consultative approach |
| • Process consultative approach |
| • Exchange of experience between companies |
| • Deeper analysis of the problem |
| • Goals and focus |

## OSH on the agenda

The opportunity to interact with new individuals during the project's workshops was perceived as a significant motivating factor for the systematic OSH management, particularly during the project's initial stages. The chance to discuss OSH matters with fellow project participants at workshops was described as enhancing OSH management commitment and prioritization. The OSH management issues raised during the workshops were addressed and processed within the respective organizations. The necessity for participating companies to take steps between the workshops placed considerable pressure on the team and the organization to deliver tangible outcomes. As the project progressed, it became evident to the participants that the work was not merely a project-specific endeavor but a crucial component of the systematic OSH management that companies were expected to address. This realization led to a shift in perspective, where the initial pressure was seen as a positive motivator.

> From being that we're just going to do something for the next meeting, it became more like: "We need to work on these issues, whether we go to a workshop or not." Initially, it took us some time, but then we booked meetings, and that's what it takes. If you book meetings, you talk about it, and they're not very long, and suddenly, you have a sustainable way of working.

(Business Area Manager, Company F)

Informants described that participation in the workshops increased attention to OSH concerns, enhanced dedication to OSH, and a higher prioritization of OSH matters. As a result of the improved knowledge base, the conceptualization of work environment issues transformed. The workshop work augmented the capacity to identify and prioritize the most pertinent work environment issues. The combination of educational theoretical elements and group work proved influential in generating new ideas and insights regarding the work environment. These ideas were then processed further at home and implemented to varying degrees. In particular, the theoretical element of OSH management was highlighted. The informants described that the work environment had become a standard meeting agenda item with accompanying minutes. They expressed hope that these new structures would result in increased implementation of work environment management in daily work.

> We think more about these issues when we work and are aware of the working environment, our protective equipment, and various other things related to our projects. On a scale of 1–10, if we were at three before, our awareness of these issues has at least gone up to five.

(Managing Director, Company C)

## The development team drives the development

The formation of the team within the project in each participating company resulted in a notable increase in the prioritization and legitimacy accorded to OSH concerns.

Establishing a distinct group convened by senior management, with the explicit purpose of addressing health and safety concerns, setting objectives, and assuming accountability while also maintaining communication and providing updates to senior management, has elevated the status of health and safety matters in relation to production targets.

*My experience is that many things can take a long time and are not prioritized, and those with responsibility have many other things to do, so they become lower on the priority list. Now, it becomes completely different when a development team works with these things, and the good structure of the work leads to effects, so we implemented many things with positive results.*

(Work Environment Coordinator, Company A)

Furthermore, the team has connected administrative and managerial roles with practical professional tasks. Following the conclusion of the project period, the teams persisted in various forms, continuing to address development issues in the domain of OSH. In some cases, existing teams were retained; in others, new teams were formed based on the model in Hälsobygget. In more nationwide companies, new teams were formed in the respective regions, as the work varies across locations, and close contact is necessary between the team and the practical activities. The participants underscored the necessity of maintaining the team to ensure continued prioritization and address work environment concerns.

*We've discussed it before: for us, it's very much about planning. We'll also develop our development teams, if you can put it that way. To begin with, we will have one for Stockholm and one in Gothenburg, which you can eventually get together to share experiences.*

(Participant during a final workshop, batch 2)

**Expert consultative approach**

The informants underscored the value of the available consultants in facilitating an expert consultative approach, which proved instrumental in ensuring a comprehensive analysis of the existing work environment issues. Of note was the consultant's expertise in designing an effective survey instrument tailored to the specific work environment issues. The survey's success hinged on the consultant's proficiency and the vital additional resources that the consultant provided. Furthermore, the informants highlighted that the consultant's external perspective enabled the formulation of questions that would not have been feasible with solely internal resources. The surveys enabled a more profound investigation of the selected work environment issue, allowing for its relevance to be substantiated or refuted.

*Yes, we could have done this, too. Then, we wouldn't have had that support and feedback. What has been most interesting for us is this measurement that has been done, which we don't have the resources and stuff to do.*

(Participant final WS, batch 1)

The consultant's ability to respond to queries, augment the body of knowledge, and proffer guidance and coaching was regarded as a valuable asset in attaining the desired outcomes for the project. The informants highlighted that the consultancy service was beneficial in addressing both specific subject issues and process-related matters. These process issues included change management, the project's direction, analyzing potential challenges and their avoidance, and implementing desired organizational changes.

*We raised the issue in our meetings of how to implement a change, and then we had many discussions about the process and potential pitfalls. We have had good training.*

(Regional Director, Company C)

## Process consultative approach

The informants emphasized that the consultants served as reflection partners for the team through a process consultative approach. They facilitated progress when the group encountered analytical and deliberative deadlocks that impeded decision-making and action. They also helped the group avoid hasty decision-making and action without sufficient analysis. The consultants' approach of eliciting the group's knowledge, posing questions to highlight alternative perspectives, and facilitating the group's decision-making process was perceived as beneficial. This approach helped the team to clarify its thoughts, identify solutions, and summarize its discussions and work.

*She's tried to help us figure things out for ourselves rather than add her knowledge, but she's drawn out our knowledge and helped us sort out our thoughts.*

(Work Environment Coordinator, Company C)

The informants highlighted the consultants' initial focus on the company's perspective and acknowledgment of the company's identification of its current work environment issues, definition of its work environment management, and confirmation of its choices as crucial aspects of the consultancy process. By adopting this approach, the consultants were able to establish a collaborative relationship with the company, creating an environment where all queries and concerns were welcomed. The consultants were instrumental in assisting the organization in establishing realistic goals in relation to the project's time frame, a contribution that was perceived as beneficial.

*They were there. They didn't hold the conductor's baton, but you had to lean on them when stuck or didn't know how to proceed, so it was an excellent support function.*

(Skilled Worker, Company E)

## Exchange of experience between companies

The exchange of experience and knowledge between the companies during the WS was perceived as interesting and instructive. Furthermore, the informants highlighted it as an essential and crucial part of the project. The significance of systematically discussing work environment issues in diverse configurations, including cross-group interactions between companies and professional groups, was underscored. The value of integrating these insights into one's group was also highlighted. The inter- and professional groups facilitated maintaining focus and prevented the discussion from reverting to the typical discourse about one's company.

*Continue to have groups with different companies, that is, a mix of other companies. Otherwise, there will probably be a lot of chatter about your company, and you will derail and start talking about something else. You can mix and sit in groups with your own company and groups with other companies. That's a great arrangement.*

(Skilled Worker, Company B)

Informants underscored the value of recognizing shared experiences and challenges across the industry, including the difficulties encountered by other companies in addressing similar issues. They identified common OSH challenges for the entire sector that transcend individual companies. The informants underscored the value of operating within the same industry, noting that they spoke the same language and had a preexisting familiarity with the business, which simplified the explanation process. The opportunity to learn from one another, receive and provide advice, and discuss potential solutions was described as a highly stimulating aspect of the process, characterized by mutual give-and-take. The exchange of experiences encompassed both work environment concerns and process-oriented issues, including cultivating employee commitment to work environment matters and disseminating messages about the work environment within the organization.

*It is very educational to listen to others and talk about their experiences, knowledge, and problems they have seen at their companies. It was very good and very educational.*

(Work Environment Coordinator, Company D)

The informants highlight that the exchange of experiences continued during coffee breaks and lunches at the workshops, with additional discussions held outside the scheduled sessions. These conversations covered various topics related to the work environment, extending beyond the immediate focus of the current workshop. Such resources might include templates and applications. The ideas and experiences exchanged during coffee breaks prompted people to consider implementing them in their organizations.

## Deeper analysis of the problems

The informants highlighted that the PBL model used during workshops provided an opportunity to halt and conduct a more thorough examination of the issues, whereby the underlying causes were subjected to a systematic and iterative investigation process. This approach proved to be a significant and transformative learning experience. The call during the workshops to actively seek out problems and learn not to be content with the initial explanation but to persist in identifying subsequent causes and their interconnections was identified as a valuable technique. By identifying the fundamental cause of the problem, it became possible to contextualize it and shift the focus to organizational issues or problems with the OSH management organization and structure. The deeper analyses of problems and causal chains conducted during the workshops counteracted the direct impulse to adopt the first proposed solution. Instead, the capacity to identify and evaluate alternative solutions was cultivated, enabling a comprehensive examination of each proposed solution's potential consequences and implications. A decision could be made only after thoroughly analyzing the problem and the proposed solution. The structure of the PBL model permitted the identification of a unifying concept within the problem formulation and the discernment of the principal issues.

*We may have become less solution-oriented and are looking at the problem instead. We're all very solution-oriented here; we would find a quick solution, then you solve the problem, and then you move on. But we have probably slowed down a bit and become a little more transparent about the issue. So, that has come out of this project.*

(Participant final WS, batch 2)

The opportunity to gain insight into the challenges and difficulties faced by other companies around the work environment during the workshops was valuable in broadening the perspective of the situation within their own company. It could become clear that there were more difficulties in the company than they had been aware of and had not given sufficient attention to. Hearing the issues presented by others resulted in a shift in perspective, prompting the generation of new

questions and the formation of a more comprehensive understanding of their own company. This process counteracted the phenomenon of "home blindness," characterized by a lack of awareness and insight into one's organization. Additionally, the composition of the team, which included not only administrative and managerial staff but also professional workers and trade union representatives, contributed to a broader perspective and approach to work environment issues, both in the management of their own company's work environment and during the workshops.

## Goals and focus

The informants underscored the significance of establishing transparent and measurable health and safety management objectives. Although the companies had previously formulated health and safety objectives, the project yielded insights that formed the basis for future initiatives. The primary points emphasized were the increased understanding of the value of having a limited number of objectives to facilitate structured progress, ensuring that each aim is clearly defined, developing a comprehensive action plan for achieving the goals, conducting regular progress monitoring in line with the plan, and maintaining group alignment. The informants underscored that the objective formulation process enabled the team to gain clarity on the desired outcomes.

*Oh my God [laughs]. It was probably during the second workshop we realized that we were going in with the wrong approach almost, with the wrong mindset, and then we redirected it at the second meeting so that we were heading in the right direction. That was the best, when we understood what we wanted.*

(Transport Manager, Company F)

## Discussion

Previous literature shows that the construction industry is one of the most dangerous sectors concerning OSH, [1–4]. SMEs account for more than two-thirds of employment in the construction industry [14], and significant challenges exist in implementing effective OSH management in SMEs overall [7–11,15,19,22]. Research on the development processes that lead to effective OSH management [19] and what strategies work for whom in what context is needed [8,22].

The aim of the study was to investigate whether and how the interventions have been beneficial in developing a well-functioning systematic OSH management. The aim was also to investigate whether and how the interventions have facilitated or impeded autonomous motivation to develop a well-functioning systematic OSH management.

Seven themes were found in the analysis: *OSH on the agenda, The development team drives development, Expert consultative approach, Process consultative approach, Exchange of experience between companies, Deeper analysis of the problems* and *Goals and focus,*

Based on the theme of *OSH on the agenda*, participation in the project and workshops required the team to address occupational safety and health (OSH) concerns raised during the workshops, process these issues within their respective organizations, take action to generate outcomes in the intervals between workshops, and report their activities and results during the workshops. This pressure was initially seen as a positive and effective motivator at the project's commencement. Such motivators can be reasonably regarded as falling within controlled motivation as identified in earlier research [41,42,68]. As the project progressed, the results indicate that the participants came to understand that the development team's efforts were not merely for the sake of project participation. Instead, these activities were undertaken because the company, by choice, recognized the importance of addressing critical issues from an OSH perspective. The described shift in the team's approach can be reasonably interpreted as transitioning from controlled motivation to developing OSH management within the company, characterized as a "must," to an autonomous motivation, conceptualized as a "we want." Previous researchers have highlighted the significance of choice as a crucial element in fostering autonomous

motivation to establish healthy and safe workplaces [69]. Participation in the workshops also appears to have fostered heightened awareness, increased dedication, and elevated prioritization of OSH issues. This phenomenon could also indicate that involvement in the project has influenced the autonomous motivation to establish effective OSH management.

Based on the theme development team drives development, the formation of a development team within the company appears to have had a decisive impact on the legitimacy of OSH issues and the assigned safety priority, defined as managers expressed concern for employees' health and safety [29,30]. Key factors in these effects appear to have been the establishment of a dedicated group convened by a senior manager, a clear focus on OSH, the setting of clear objectives, the follow-up on these objectives and the team's described role in establishing a more explicit link between practical work, management, and administration regarding OSH. Considering the composition of the team with administrative and managerial staff, professional workers, and union representatives, and its role in conducting work environment activities within the project, with limited consultant support, the described key factors indicate employee participation in planning the company-specific project and the emergence of joint ownership of problems and solutions. Previous research has identified employee involvement as a decisive factor in the effectiveness of interventions aimed at improving occupational safety and health [33,34]. Additionally, it has been found that steering groups need to include both employees and managers [34] to effectively implement relevant OSH interventions. As previous research has indicated, the concept of assigned safety priority has been recognized as a core dimension of safety climate [29], defined as managers expressed concern and clear actions supporting employees' health, safety, and well-being [30]. Leadership commitment to health and safety has been shown to influence employee behavior, thereby contributing to the development of a more favorable safety climate [30,38,70]. Managers' dedication to health and safety has been recognized as a significant factor influencing the safety climate, alongside employee participation and involvement in health and safety issues [33,71–74]. Cautiously, the results may suggest that the composition of the development team in terms of professions and the team's opportunity to report to senior management have contributed to elevating the status of OSH relative to production targets, implying an increased assigned safety priority. The teams appear to have been perceived as functional and essential in developing OSH management, as all participating companies state that they will maintain, develop, and, in some cases, create more teams within the company to continue addressing development issues in OSH. The team were also considered as necessary to ensure that work environment concerns remain prioritized and addressed, i.e., maintaining a high level of assigned safety priority. From a motivational perspective, a reasonable interpretation of the results is that the team has enhanced collective efficacy [32] in addressing OSH concerns. Additionally, from an SDT perspective, it appears that the team has somewhat satisfied the psychological need for relatedness, thereby suggesting that a consequence of the team is enhanced autonomous motivation for OSH management.

Based on the theme of an expert consultative approach, this form of guidance was seen as helpful mainly in two ways. First, by providing practical assistance in designing and offering an additional resource—conducting surveys that the companies involved likely couldn't have done on their own. Second, by answering specific questions on environmental-related topics and process-oriented areas, thus expanding the team's knowledge base. According to the principles of SDT [31,75], autonomous motivation can be enhanced by supporting the three fundamental psychological needs of competence, relatedness, and autonomy. Knowledge and experience are factors that can be assumed to positively influence the psychological need for competence [75,76], which, in this context, can also be presumed to positively influence autonomous motivation toward OSH management.

Based on the theme *process consultative approach*, the focus on the company's perspective, the respect and affirmation for the company's prioritization of work environment issues, and the alignment with the company's definition of work environment management were identified by the participants as essential elements for establishing functional cooperation and creating a climate of trust. This finding aligns with the conclusions of previous scholars [52–54] who posited that when process consultation is employed, the consultant and the client are equal partners and the consultant's role is to assist the client to take ownership of their problems, maintaining autonomy in their own initiatives, formulating their own diagnoses,

and devising suitable and functional interventions. To act as a reflection partner with a motivational guiding style, adopted from Motivational Interviewing [35], for the team seems to have been perceived as beneficial in several ways. By eliciting the team's knowledge, asking questions about alternative perspectives, and assisting the team in avoiding hasty decisions without sufficient analysis, the process consultative approach appears to have contributed to the team's enhanced ability to articulate their thoughts, identify solutions, and synthesize discussions and work. This would be in line with the central assumption in process consultation that "one can only help a human system to help itself" [52, p.1] When viewed through the motivational lens of SDT [31,75], those findings suggest that the process consultative approach may have contributed to satisfying the psychological needs for autonomy and relatedness. Furthermore, when the two consultative approaches are considered together, they seem to positively impact the psychological need for competence [31,75]. Thus, it can be suggested that both consultative approaches may have contributed to a greater degree of autonomous motivation for OSH management.

Based on the theme of *exchange of experience between companies*, participation in the project through workshops that provided the opportunity to collaborate and exchange experiences and insights about the domain of the occupational environment with representatives of other companies was highlighted as the most beneficial intervention. In the exchange of experiences, learning that others are grappling with analogous challenges was also emphasized as beneficial. The current intervention project was partly based on a prior network model in which multiple companies collaborated to implement systematic OSH management in their respective entities [27]. This collaboration transpired through participation in a series of structured and themed network meetings, led by a supervisor, with the other participating companies. In the preceding evaluation research [27], the opportunity to exchange experiences and materials, in conjunction with the commitment to allocate time for work environment management, were identified as the most beneficial factors in the network model. The realization that there are common challenges in the construction industry work environment seems to have had a relieving effect, as it demonstrated that one is not alone in facing work environment issues that are difficult to overcome. Furthermore, recognizing that many of these challenges are not unique to a single company but are widespread across the construction industry also seems to have the same effect. From a motivational perspective, it can be assumed that realizing that other companies face similar work environment problems and challenges within the construction industry can contribute to satisfying the psychological needs for relatedness [31,41,68] to a greater extent. This, in turn, can lead to increased autonomous motivation [77] to engage in occupational safety and health.

Based on the theme of a *deeper analysis of the problem*, using a PBL model [36,58] during workshops seems to have been a significant learning experience that contributed to the participants' increased ability to conduct more profound analyses of the OSH problems they identified. Using the steps in the model provided an opportunity to actively look for OSH problems, to stop and systematically examine the identified OSH problem, to take the time to look for causal chains and root causes, and to place the identified problem in its own organizational and structural context. The process described is consistent with suggestions from previous research [58], which underscores the necessity of incorporating steps to facilitate continuous reflection, the implementation of corrective measures, and the integration of learning in PBL processes within SMEs. The exchange of experiences and insights seems to have facilitated the acquisition of novel perspectives on one's own company's work environment challenges and the identification of alternative solutions, in addition to the generosity of other participants, who proffered tips, information, and ideas. According to O'Brien et al. [58], previous research has indicated that implementing a PBL methodology in SMEs is supported by creating a network of peers, where both formal and informal learning methods are employed. A cautious interpretation is that the exchange of experiences and insights through participation in the project has played a role in this process. Additionally, a cautious interpretation suggests that this exchange in question has positively affected the participants themselves, as indicated in previous research [58]. This interpretation suggests that the exchange has helped promote a responsible approach to identifying relevant problems that can serve as a trigger for a PBL process.

The concept of collective efficacy, defined as the group's ability within an organization to undertake and successfully complete a developmental task [32], is similar to the concept of self-efficacy but at a collective level. According to existing research, students' self-efficacy appears to be positively influenced when they have opportunities to actively participate and contribute to learning [78]. Even if the present study is in a different context, a reasonable assumption may be that the described learning experience in the project can have contributed to increased collective efficacy [32] and increased the sense of competence, which is also considered to contribute to increased autonomous motivation [77].

The theme's *goals and focus* indicated that participating in the project yielded an increased awareness of the importance of goal clarification for creating structure in the team's and organization's OSH work, clarifying the desired outcomes, and ensuring team alignment. This aligns with what has been suggested in previous research [79], that establishing clear goals is a prerequisite for developing effective and well-functioning teams at work. In the domain of motivation, earlier research indicates that specific goals tend to generate a more pronounced motivational effect compared to non-specific goals [80].

## Limitations and future research

There are some limitations to the study. The first limitation is the absence of long-term follow-up regarding elevated assigned safety priority outcomes and augmented engagement with OSH management initiatives. Consequently, the long-term effects of project participation and interventions remain uncertain. A second limitation is that only those participating in the teams have been interviewed. Therefore, it is unclear how the commitment to OSH management has been disseminated throughout the organization and what the general perception of the assigned safety priority or OSH management is within the organization. As the project did not have control over the choice of the team members, a third limitation is that those included in the teams may already have high commitment and autonomous motivation regarding OSH issues. Therefore, the results may have been biased towards high-priority safety assignments and autonomous motivation for OSH issues. Future research would benefit from combining qualitative studies of intervention effects among intervention participants with quantitative analyses of the diffusion of intervention effects across the organization.

## Conclusions

The aim of the study was to investigate whether and how the interventions have been beneficial in developing a well-functioning systematic OSH management. The aim was also to investigate whether and how the interventions have facilitated or impeded autonomous motivation to develop a well-functioning systematic OSH management.

The overall results of the interviews indicate that the assigned safety priority has increased within the participating companies. Additionally, there are indications that a willingness has developed among the companies to continue and strengthen their efforts in addressing urgent OSH issues in a structured manner. This also includes ongoing efforts to develop an effective and well-functioning systematic OSH management system. These results could likely stem from an increased autonomous motivation to create a healthy and safe work environment for all employees. In light of the motivational theories of Self-Determination Theory [31] and collective efficacy [32], the interpretations of the motivational influencing functions of the various interventions are as follows. The collective efficacy [32] of the participating companies, and the psychological need for competence among participants, appears to have been positively impacted by their involvement in workshops, which included problem-based learning [36,37,58], exchange of experience and knowledge, the work and structure of the team, and both process and expert consultation. The psychological need for relatedness appears to have been primarily positively influenced by participation in workshops, work within the development team, and experiencing a process consultative approach [53,54]. The psychological need for autonomy seems to have been positively influenced by exposure to a process consultative approach. Based on the results of the study, some practical implications for creating greater commitment and engagement in OSH management in structured intervention projects may be relevant. Interventions and the establishment of a dedicated development team can benefit from being organized

so that participants experience autonomy and ownership of the selected OSH area and their work processes within the team as early as possible. Since sharing experiences and knowledge, both structured and informal, appears to have been an important factor in developing collective efficacy and enhancing feelings of competence and belonging, future OSH intervention projects might benefit from dedicating more time and resources to these activities. Future OSH intervention projects, for instance, within occupational health services, may become more cost-effective by focusing external resources on a process-oriented consultative approach and exclusively providing expert consultation when needs are identified and clearly expressed.

## Author contributions

**Conceptualization:** Niklas Rydbo.

**Formal analysis:** Niklas Rydbo, Kristina Aurelius, Martin Grill, Christian Jacobsson, Anders Pousette.

**Funding acquisition:** Anders Pousette.

**Investigation:** Niklas Rydbo, Kristina Aurelius, Martin Grill, Anders Pousette.

**Methodology:** Niklas Rydbo, Kristina Aurelius, Martin Grill, Christian Jacobsson, Anders Pousette.

**Supervision:** Christian Jacobsson, Anders Pousette.

**Writing – original draft:** Niklas Rydbo, Kristina Aurelius, Martin Grill, Christian Jacobsson, Anders Pousette.

**Writing – review & editing:** Niklas Rydbo, Kristina Aurelius, Martin Grill, Christian Jacobsson, Anders Pousette.

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
