## [Decision Letter · Decision Letter 0]

15 Jan 2025

Dear Dr. Rydbo,

Thank you for submitting your manuscript to PLOS ONE. After careful consideration, we feel that it has merit but does not fully meet PLOS ONE’s publication criteria as it currently stands. Therefore, we invite you to submit a revised version of the manuscript that addresses the points raised during the review process.

Please note that we have only been able to secure a single reviewer to assess your manuscript. We are issuing a decision on your manuscript at this point to prevent further delays in the evaluation of your manuscript. Please be aware that the editor who handles your revised manuscript might find it necessary to invite additional reviewers to assess this work once the revised manuscript is submitted. However, we will aim to proceed on the basis of this single review if possible.

We look forward to receiving your revised manuscript.

Kind regards,

Avanti Dey, PhD

Staff Editor

PLOS ONE

Journal Requirements:

2. Thank you for stating the following financial disclosure: This work was supported by AFA Insurance (grant number 160151 ) and Swedish Transport Administration (grant number TRV2016/107905) and Region Västra Götaland, Occupational and Environmental Medicine (in-kind financing)

Reviewers' comments:

Reviewer's Responses to Questions

**Comments to the Author**

1. Is the manuscript technically sound, and do the data support the conclusions?

Reviewer #1: Partly

2. Has the statistical analysis been performed appropriately and rigorously?

Reviewer #1: No

3. Have the authors made all data underlying the findings in their manuscript fully available?

Reviewer #1: Yes

4. Is the manuscript presented in an intelligible fashion and written in standard English?

Reviewer #1: No

Reviewer #1: Please consider the attached review points.

(*) Overall, this document should be worked on to make necessary adjustments regarding academic writing/phrasing.

85, 94, 197 – Authors should avoid using “we,” “I,” or “our”, etc. in communicating the research process or findings.

97, 98 – You need to systematically introduce the elements of a participative intervention in this study section before indicating the purpose of examining how they foster a well-functioning and continuous SWEM in small and medium-sized construction. This will help create a basis for examining the elements of a participative intervention,

198, etc. – Check your first mention of DTs and others to see if the actual words were indicated before stating acronyms. Please check the manuscript thoroughly and amend it accordingly. Besides, you need to reduce the use of acronyms significantly.

243 – 246 – Your aim can be combined and indicated in a paragraph as against in bullets.

Methods

Authors should indicate how they generated their interview questions.

You have not indicated a process of Themes. What are your workable themes? Hence, how did you decide to implement a semantic inductive thematic analysis?

This section needs some professional layout. It is not robust and procedural, which is the highlight of a qualitative research methodology.

Results

As indicated earlier, you did not indicate your working themes. Hence, reporting interviews in themes you did not indicate is strange.

Besides, all reported statements of respondents (interviewees) should be in italics.

Discussion

Further to your discussions, authors should systematically infer their findings under each theme indicated and relate them to previous scholars’ conclusions.

This will serve as the achievement of the aim of the study about the themes.

Conclusions

Furthermore, your conclusion should start with the aim of the investigation, followed by a summary of your results.

Besides, this section indicates the implications of your findings to practice. This could further generate some suggestions.

References

Please cross-check your reference list to ensure they are in the correct format. Besides, check the reference list with your in-text citations.

**Do you want your identity to be public for this peer review?** For information about this choice, including consent withdrawal, please see our Privacy Policy

Reviewer #1: No

---

## [Author Response · Author response to Decision Letter 1]

20 Aug 2025

Journal requirements

1.As we understand the instructions, the manuscript complies with PLOS ONE’s style requirement. We have not been able to find instructions on how files should be named, so it is possible that the file names may need to be corrected. If so, we apologize for this.

2. Founders had no role and this has been stated in te revised cover letter

3. The restrictions have been changed to "available upon request," which has been stated in the text. An explanation is provided about ethical restrictions

Reviewer comments

1. The manuscript has been carefully revised

2. The section on methodology has been expanded and supplemented based on qualitative methodology.

3. -

4. The manuscript has been revised in terms of language

5.The manuscript has been revised based on the reviewers comments. The changes have been described in the attached "response to reviewers".

---

## [Decision Letter · Decision Letter 1]

10 Sep 2025

Building health: motivational factors to enhance systematic work environment management in the construction industry

PONE-D-24-50145R1

Dear Dr. Rydbo,

We’re pleased to inform you that your manuscript has been judged scientifically suitable for publication and will be formally accepted for publication once it meets all outstanding technical requirements.

Kind regards,

Henri Tilga, PhD

Academic Editor

PLOS ONE

Additional Editor Comments (optional):

Reviewer #1:

Reviewers' comments:

Reviewer's Responses to Questions

**Comments to the Author**

Reviewer #1: All comments have been addressed

2. Is the manuscript technically sound, and do the data support the conclusions?

Reviewer #1: Yes

3. Has the statistical analysis been performed appropriately and rigorously?

Reviewer #1: Yes

4. Have the authors made all data underlying the findings in their manuscript fully available?

Reviewer #1: Yes

5. Is the manuscript presented in an intelligible fashion and written in standard English?

Reviewer #1: Yes

Reviewer #1: Thanks to the authors for attending to the recommendations. However, you might still need to look thoroughly into the manuscript to perfect some grammatical structuring, spellings, and ideas.

**Do you want your identity to be public for this peer review?** For information about this choice, including consent withdrawal, please see our Privacy Policy

Reviewer #1: No

---

## [Editor Report · Acceptance letter]

PONE-D-24-50145R1

PLOS ONE

Dear Dr. Rydbo,

I'm pleased to inform you that your manuscript has been deemed suitable for publication in PLOS ONE. Congratulations! Your manuscript is now being handed over to our production team.

Kind regards,

on behalf of

Dr. Henri Tilga

Academic Editor

PLOS ONE